# Segregation of liquid crystal mixtures in topological defects

Mohammad Rahimi[1], Hadi Ramezani-Dakhel[1], Rui Zhang[1], Abelardo Ramirez-Hernandez[1,2], Nicholas L. Abbott[3] & Juan J. de Pablo[1,2]

The structure and physical properties of liquid crystal (LC) mixtures are a function of composition, and small changes can have pronounced effects on observables, such as phase-transition temperatures. Traditionally, LC mixtures have been assumed to be compositionally homogenous. The results of chemically detailed simulations presented here show that this is not the case; pronounced deviations of the local order from that observed in the bulk at defects and interfaces lead to significant compositional segregation effects. More specifically, two disclination lines are stabilized in this work by introducing into a nematic liquid crystal mixture a cylindrical body that exhibits perpendicular anchoring. It is found that the local composition deviates considerably from that of the bulk at the interface with the cylinder and in the defects, thereby suggesting new assembly and synthetic strategies that may capitalize on the unusual molecular environment provided by liquid crystal mixtures.

[1] Institute for Molecular Engineering, University of Chicago, Chicago, Illinois 60637, USA. [2] Argonne National Laboratory, Argonne, Illinois 60439, USA. [3] Department of Chemical and Biological Engineering, University of Wisconsin–Madison, Madison, Wisconsin 53706, USA. Correspondence and requests for materials should be addressed to J.J.d.P. (email: depablo@uchicago.edu).

Topological defects[1], which arise from symmetry breaking considerations, have a significant influence on the macroscopic properties of materials. Liquid crystals (LCs) exhibit a rich variety of defects that can be easily observed and even controlled experimentally. Such defects are closely analogous to those encountered in cosmology, particle physics and condensed matter physics, thereby making LCs a valuable experimental testing grounds for the study of defects in a wide range of disciplines[2]. As such, they have long been the subject of considerable interest[3–20], particularly given the fact that many of the features that make LCs exciting for emerging applications can be traced back to their existence. In sensors, for example, the interaction of lipids with LC defects provides the basis for ultra-high sensitivity[21]. In chiral LC materials, the structure of the defects provides the basis for fast electro-optical switching[22]. In materials design, nematic-driven particle self-assembly enables creation of organized particle clusters whose nature depends on the morphology of LC defects[23,24], and it was shown that the structure of the defects can trigger processes of molecular self-assembly[25,26]. In contrast, in display technologies they represent important obstacles for light transmission[27].

Nematic LCs are anisotropic fluids that are aligned along the so-called nematic director $\mathbf{n}$. The equilibrium nematic director can be determined by minimizing the Frank elastic free energy, leading to a uniform nematic director field for the bulk[28]. The interaction between LC molecules and a surface, which is referred to as anchoring, can distort the overall nematic director, serving to underscore the delicate balance of elastic and surface contributions to the free energy that governs the behaviour of these materials. Indeed, director distortions can be driven by confinement[29] or by addition of particles[30]. In many cases, as a result of these distortions, the degree of local orientation in specific regions is reduced and the director field can no longer be uniquely defined, leading to the formation of discontinuities. The LC defects correspond to these regions of low order. In continuum descriptions of LC materials, such as those introduced by Landau and de Gennes, a so-called $\mathbf{Q}$ tensor is introduced to avoid the mathematical challenges associated with such discontinuities[28]. Defects are commonly classified according to their topological charge, $M$, which is either $\pm 1$ or $\pm \frac{1}{2}$ for a nematic LC. The magnitude and sign represent, respectively, the number and the direction of the nematic field rotation in a complete defect circuit[28]. The interactions that arise between defects are in many respects similar to those encountered in electrostatics; LC defects of the same sign repel each other, while those having opposite sign attract each other[5]. The total topological charge is constrained in any given system. For example, if a spherical particle with strong homeotropic anchoring carries with it a hedgehog defect with topological charge of $+1$, in a bulk nematic LC, whose charge was originally zero, the defect increases the total topological charge of the system to unity. In that case, additional defects must therefore be generated in order to satisfy the total topological constraint of zero charge. This can be achieved in one of two ways: by forming elsewhere a point defect with a topological charge of $-1$ (ref. 31), or by forming a line defect around the spherical particle, a so-called Saturn ring defect[32]. Indeed, the Saturn ring actually consists of two disclination lines with a topological charge of $-\frac{1}{2}$ that are bended around the spherical particle. Saturn rings are stable only for nano-sized particles[33]. Alternatively, when a cylinder with homeotropic anchoring is introduced, from a topological point of view two types of defects are possible: a disclination line with a topological charge of $-1$, or a pair of disclination lines with topological charge of $-\frac{1}{2}$. It turns out that a pair of disclination lines is more favourable from an energetic perspective[34–36].

As the arguments above have all been generated on the basis of phenomenological descriptions of LCs, they have been verified by a wide range of experimental observations. Such arguments, however, view defects as a region of space where the director changes abruptly. The molecular details of how that happens, however, remain a mystery. One of the questions that we seek to address is whether, for LC mixtures, the distortion of nematic director has an influence on the composition. Note that LC mixtures are widely used in commercial applications, and their physical properties depend strongly on composition, with small changes being able to compromise the isotropic to nematic phase. Traditional treatment of LCs, such as the Landau–de Gennes theory, have generally assumed that LC mixtures are compositionally homogeneous even at the core of the defect.

In this work, we employ molecular dynamics (MD) simulations to characterize the structure of the LC defects at atomic length scales. A cylinder with strong homeotropic anchoring is created in the centre of the simulated system by applying an external repulsive potential, leading to the formation of two disclination line defects that run parallel to the cylinder. Past studies of line defects along a cylinder have been limited to Landau–de Gennes continuum analyses[34,35], Lebwohl–Lasher-type lattice models[37], and highly coarse-grained Gay–Berne representations that assume molecules to consist of smooth, uncharged ellipsoidal particles[38]. To the best of our knowledge, line defects have not been analysed with atomic-level resolution. By simulating a mixture of 5CB (4-cyano-4′-pentylbiphenyl) and 8CB (4-cyano-4′-octylbiphenyl), we present a previously unknown molecular view of the scalar order parameter, density, composition, biaxiality and the nematic director field inside line defects of what is perhaps the most widely studied liquid crystalline material.

## Results

**Models and setup.** A standard simulation protocol at the united-atom level of description is adopted to simulate a 5CB/8CB mixture consisting of 8,000 5CB and 8,000 8CB molecules. The simulation is carried out at a pressure of 1 atm and a temperature of 300 K. Note that 5CB and 5CB/8CB mixtures exhibit a nematic phase under these conditions, while 8CB is in the smectic phase. The 5CB/8CB mixture is fully equilibrated in a cuboid simulation box with periodic boundary conditions and having dimensions $31 \times 31 \times 7$ nm$^3$. A neutral cylinder with a soft boundary is created by applying a radial force to all atoms located in the cylinder. The cylinder is formed at the centre of the simulation box along the Z axis, and has a radius of 5 nm (Fig. 1a). Note that the radial force is a function of distance from the centre of the cylinder, and it reaches zero at the cylinder's surface.

**Homeotropic cylinder.** Our analysis begins with an examination of the orientation of LC molecules at the cylinder's surface. We quantify the orientation of the molecules located within a cylindrical shell around the cylinder of thickness of 3 nm and determine the surface anchoring. Specifically, we calculate the second-order Legendre polynomials $P_2(\cos\theta) = \left\langle \frac{3}{2}\cos\theta^2 - \frac{1}{2} \right\rangle$, where $\theta$ is the angle between the long molecular axis and the surface normal, and $\langle \ldots \rangle$ denotes an average over all molecules located in the cylindrical shell over time. The value of $P_2$ represents the type of anchoring: a positive value corresponds to homeotropic anchoring, and a negative value corresponds to planar anchoring. For the cylinder considered here, $P_2$ is positive, $P_2 = 0.61$, indicating that the cylinder exhibits homeotropic anchoring. To further study the orientation of the LC molecules at the interface, we calculate the distribution of $\cos\theta$ as a function of distance from the cylinder's surface. We observe two sharp

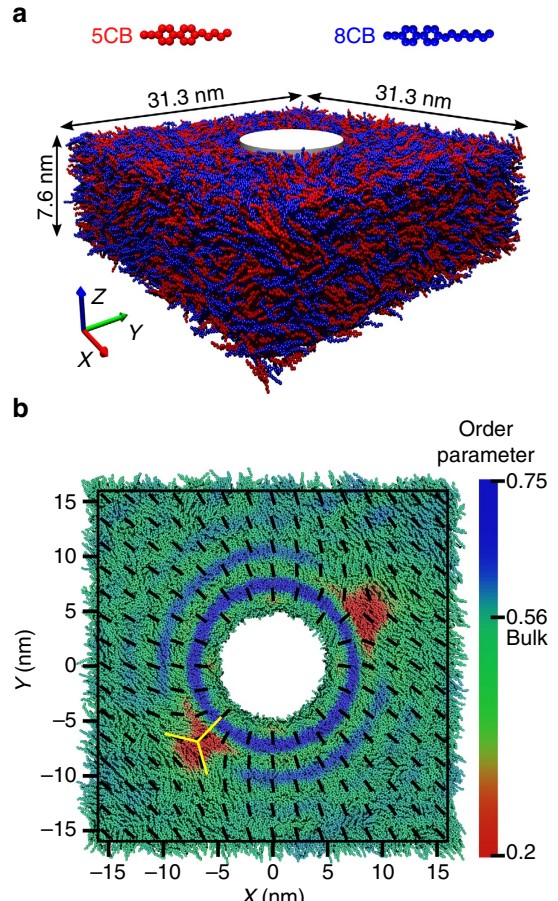

**Figure 1 | Top view and side view of the simulation box with two line defects.** (**a**) Configuration of the simulation box including a cylindrical hole with radius 5 nm at the centre. (**b**) Top view of the simulation box. The colours represent the local scalar order parameter and the black lines show the direction of the nematic field. The homeotropic cylinder disturbs the nematic field and forms two topological line defects with topological charge − ½. At the defects, molecules are disordered, leading to the low value of the order parameter.

peaks at $\cos\theta = 1$ and $\cos\theta = -1$, corresponding to an anti-parallel arrangement of LC molecules at the interface (Supplementary Note 5 and Supplementary Fig. 7). This anti-parallel arrangement, leading to homeotropic anchoring, has also been reported in experiments at free interfaces[39] and in past numerical calculations of such interfaces[40]. At a free interface, there are two competing contributions to the surface anchoring; short-range intermolecular interactions, which lead to planar anchoring and long-range intermolecular interactions (electro-static interactions), which promote homeotropic anchoring[41]. For LC molecules with large longitudinal dipoles, such as 5CB and 8CB, the long-range interactions dominate short-range forces and, therefore, a free surface induces homeotropic anchoring[42,43]. In our model, in analogy to the effects of a free surface, long-range intermolecular interactions force the LC molecules into a perpendicular antiparallel arrangement at the cylinder's surface. Note that the antiparallel arrangement is also observed in the bulk of the 5CB/8CB mixture (Supplementary Note 4 and Supplementary Fig. 5).

As alluded to earlier, the distortion of the nematic director arises from the balance of elastic and surface energy contributions to the free energy. For a particle with radius $R$, the elastic energy scales as $\propto KR$, where $K$ is an elastic constant that quantifies deformations of the LC, and the surface energy scales as $\propto WR^2$,

where $W$ is the anchoring strength. These scaling arguments indicate that only particles that are comparable or larger than the so-called surface extrapolation length, $\xi = K/W$, can perturb the nematic field and generate defects. In order to determine the surface extrapolation length in our model, we first calculate the surface anchoring strength as a function of distance from the interface. To do so, we map the surface free energy obtained from our simulations into a Rapini–Papoular expression (Supplementary Note 6 and Supplementary Fig. 8); we find values for $W$ in the range between $5.0 \times 10^{-3}$ and $3.5 \times 10^{-2}$ $\mathrm{J\,m^{-2}}$ within 3 nm distance from the interface. Note that the anchoring strength depends on surface curvature, since lateral molecular packing on a flat surface is likely to be more efficient, leading to stronger ordering. The elastic constants reported for the molecular model adopted here are 5 and 10 pN for 5CB and 8CB, respectively[44,45]. For a 5CB/8CB mixture, that constant should be in the range between 5 and 10 pN. Thus the surface extrapolation length for the model considered here is in the range from 0.15 to 2 nm, which is smaller than the radius of the cylinder (5 nm). The cylinder considered here should therefore be able to induce formation of well-defined line defects, which is precisely what we find in our simulations.

**Local order parameter and nematic field**. To examine the orientation of the nematic director in such line defects, we cal-culate all components of the **Q** tensor. Such a tensor is defined as a spatial average of the dyadic product of individual molecular orientations. The scalar order parameter is the largest eigenvalue of the **Q** tensor and the corresponding eigenvector is the nematic director **n**. In physical terms, the ensemble average of molecular orientations within a given volume is the nematic director, and the scalar order parameter quantifies the extent of orientation along the nematic director. The possible range of values for $S$ is between 0 and 1.0: $S = 0$ corresponds to disordered states, while $S = 1.0$ represents perfect order. In order to obtain a local value of the **Q** tensor, we divide the simulation system into small bins, and calculate the components of **Q** in each bin by averaging over all molecules present in the bin. Our calculations are first performed using cuboid bins, whose length in the $Z$ direction spans the entire simulation system (Supplementary Fig. 3). For reference, our analysis begins with an equilibrated system without the cylinder. The scalar order parameter is uniform in the whole simulation box with average $0.56 \pm 0.06$, and the nematic director field is oriented along the diagonal of the $XY$ plane (Supplementary Note 3 and Supplementary Fig. 4). Note that molecules are free to orient and align in other directions without experiencing an energy penalty. As a next step, we apply the radial force and generate the homeotropic cylinder shown in Fig. 1b (top view of the simulation box). Colours represent the value of the scalar order parameter and black lines display the direction of the nematic field. As shown in the figure, two defects having a low scalar order parameter are clearly visible in the vicinity of the cylinder's surface. Around the cylinder, the nematic director is perpendicular to the surface, confirming that the cylinder induces homeotropic anchoring. The nematic field ori-ents smoothly as a function of distance from the cylinder surface and stands along the diagonal of the $XY$ plane at the edge of the simulation box. It is only at the defects that the nematic director experiences a discontinuity, as revealed by the low scalar order parameter. Rotation of the nematic filed around the defects indicates that they have a − ½ topological charge. Given the symmetry of the cylinder, we rely on a polar coordinate system to analyse the structure. Figure 2a shows the scalar order parameter profile in such a system, where cylindrical shells around the cylinder are divided into small bins (Supplementary Fig. 3). The

presence of two defects with low scalar order parameter is more pronounced in Fig. 2a.

The biaxiality $\mu$ is often used to characterize LC defects. Theoretical calculations predict the existence of biaxial domains around and at the core of defects[10]. Breaking the symmetry of molecular orientation along the nematic director orients the molecules along the biaxial direction, which is perpendicular to the nematic director. The biaxiality quantifies the degree of orientation along the biaxial direction and is given by the second-largest eigenvalue of the **Q** tensor. Figure 2b shows the calculated biaxiality extracted from the **Q** tensor in polar coordinates. One can appreciate that $\mu$ is zero everywhere, indicating that the system is uniaxial, except for small regions of space localized at the defects' cores. Their size, which is comparable to that of the defects, is a few biaxial coherence lengths, 3.4 nm for 5CB at room temperature[46]. Varying director patterns close to the defects are responsible for the biaxial domains: they are observed whenever the symmetric tensor $\mathbf{M} = (\nabla \mathbf{n})(\nabla \mathbf{n})^T$ has two non-zero eigenvalues[47]. This observation is particularly interesting, as within the Landau–de Gennes theory the stable state is either isotropic or uniaxial and biaxiality is observed for intrinsically uniaxial nematic LCs, based on the definition of the **Q** tensor. Note that experiments with colloids are far from being able to verify these predictions due to the small size of the defects, and a biaxial domain has only been observed in a thin hybrid channel using atomic force microscopy[46]. Our findings, however, clearly support past theoretical predictions, and a comparison of Fig. 2a,b indicates that the position and the size of the regions with low scalar order parameter—the defects—are the same as those of the biaxial domains.

Figure 2a also shows the existence of high-order regions that follow an oscillatory pattern. Specifically, the scalar order parameter oscillates around the bulk value as a function of the distance from the cylinder surface, and in the defects, such oscillations are damped. The defects are located right at the first peak, and they attenuate the second and third peaks. In contrast, the oscillations continue in other directions over length scales of almost 10 nm from the surface. At least three maxima are visible at 1.6, 4.3 and 7.1 nm from the surface; beyond the latter distance, the scalar order parameter converges to the bulk value. The density profile shown in Fig. 3a shows the same patterns that are apparent in Fig. 2a, indicating that LC molecules form organized layers around the cylinder surface. The number and the frequency of the maxima in the density profile are identical to those observed in the scalar order parameter profile. Of course the layers are not uniform due to the presence of the defects. Figures 2a and 3a show that the scalar order parameter and density oscillations are damped at the defect, as ordered molecular

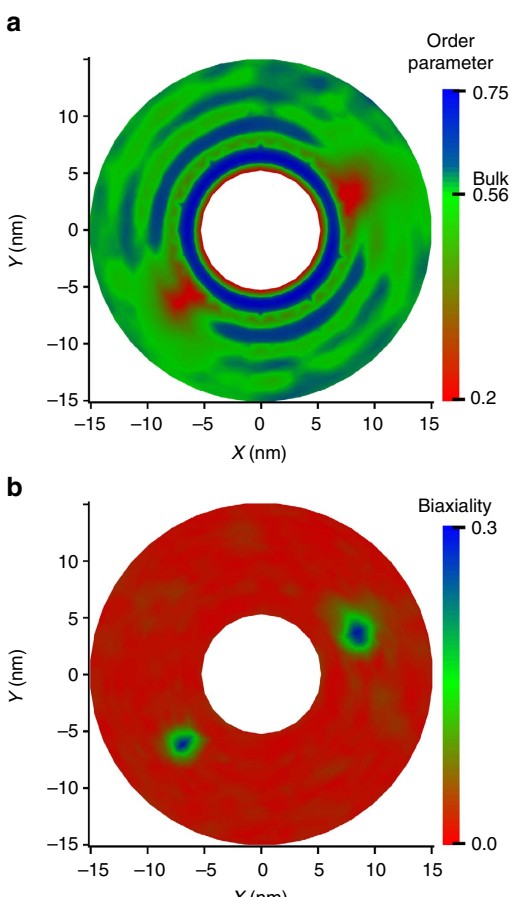

**Figure 2 | The local order parameter and biaxiality. (a)** The local scalar order parameter in a polar coordinate system shows two disclination line defects that run parallel to the cylinder. The interface induces positional and orientational order, forming smectic layers around the cylinder. The second and third smectic layers are collapsed at the location of the defects. **(b)** The biaxiality map shows two biaxial domains localized at the defect cores.

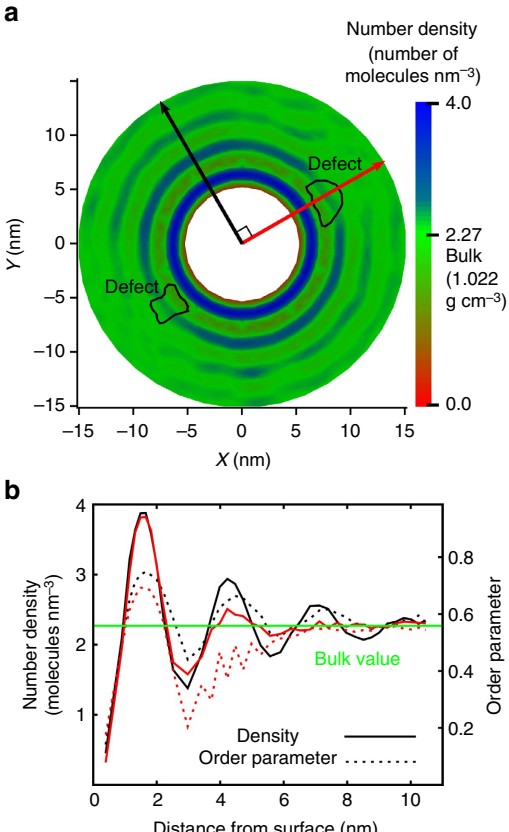

**Figure 3 | Order parameter and number density into a defect. (a)** The density profile shows oscillations around the cylinder. **(b)** The density profile and scalar order parameter as a function of distance from the cylinder's surface, plotted along two directions, is shown by the two arrows in **a**. The red arrow goes through the defect cores, and the black arrow is perpendicular to the red arrow. The green line corresponds to the bulk value of the density, namely 2.27 molecules nm$^{-3}$ (1.022 g cm$^{-3}$) and order parameter 0.56. In the black direction, we observe damped oscillations in order parameter and density having similar frequency. In the red direction, the defect damps the oscillations after the second maximum.
The magnitude of the standard deviations for these computed values is less than 2% in both directions.

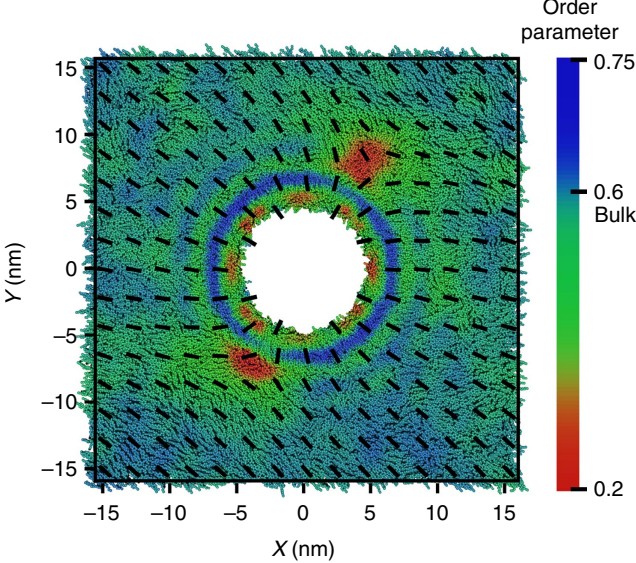

**Figure 4 | Configuration of two line defects in pure 5CB.** Top view of the pure 5CB simulation box at 295 K. The figure shows a colour map of the local scalar order parameter. The black lines represent the direction of the nematic field. The homeotropic cylinder breaks the symmetry of the system and leads to formation of two topological line defects where the molecules are disordered, leading to a low order parameter.

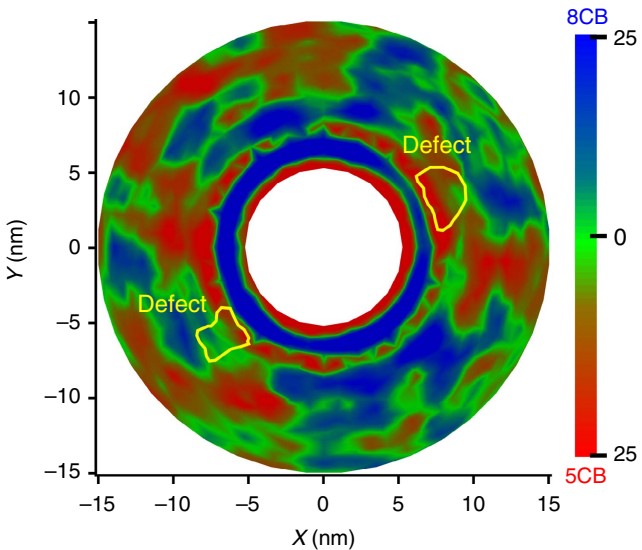

**Figure 5 | Segregation of 5CB/8CB mixture.** The composition profile shows the concentration of 8CB and 5CB molecules in the system. It is given by $\frac{\rho_{8CB} - \rho_{5CB}}{\rho_{8CB} + \rho_{5CB}} \times 100$, where $\rho_{8CB}$ and $\rho_{5CB}$ are the 8CB and 5CB density, respectively. The concentration of 8CB molecules in one layer around the cylinder is around 25% higher than that of 5CB. This layer is followed by a layer with higher concentration of 5CB molecules.

arrangements cannot persist. The density profiles as a function of distance from the surface along two different directions in Fig. 3b show that, at the defect, the density is lower than in other regions, even when those are located at the same distance from the surface. This finding is consistent with theoretical calculations provided in Supplementary Note 10, which show that an increase of the Landau–de Gennes free energy at the defect is responsible for the deviations of density. It is important to note that a nanometer interfacial region with high-order parameter was also observed in simulations of 5CB at a silicon surface with planar anchoring[48]. The authors reported that the silicon surface induces orientational and positional order, leading to formation of a high-order region at the interface, consistent with our observations at the cylinder surface.

In nanocomposite materials, for example, it is well-known that the interaction between nanoparticles and a polymeric matrix induces molecules to form layers around the nanoparticles[49,50]. For the case of rod-like molecules, the only way to pack them into a high-density layer is to align them along a specific direction, which inevitably leads to correlations between the scalar order parameter and the density. Strong homeotropic anchoring at the cylinder induces ordering, leading to the first peak in the scalar order parameter and the density profiles. The first peak exceeds the bulk density and the scalar order parameter by factors of 1.7 and 1.4, respectively. The first layer is composed of pairs of antiparallel molecules around the cylinder. As noted earlier, at the interface, dipole–dipole interactions between two molecules align them in a direction such that the polar heads are closer together than the tails (Supplementary Fig. 6). This antiparallel arrangement of the two dipoles forms a quadrupole. This structure is energetically favourable, so much so that molecules at the interface are ordered even at temperatures above the nematic–isotropic transition (Supplementary Note 7 and Supplementary Fig. 9). This observation is consistent with experimental observations of 5CB at a cavity wall, where the surface induces ordering of the first molecular layer at temperatures well into the isotropic phase[51]. It is important to note that this behaviour cannot be explained by the Landau–de

Gennes continuum theory. In fact, the first layer is smectic, as the molecules possess both orientational and positional order. It is well known that pure 8CB molecules are organized into smectic layers with antiparallel arrangement at 300 K (ref. 52), while pure 5CB molecules are in the nematic phase at the same temperature[53]. According to our calculations, the 5CB/8CB mixture is in the nematic phase (Supplementary Fig. 1); the surface induces orientational and positional order to the first layer and generates a smectic region, whose structure propagates well into the bulk region. Note that this finding is consistent with experimental observations of 8CB at a free surface, where surface smectic layers appear at a temperature above the smectic–nematic transition temperature[54]. The distance between layers is approximately 2.7 nm, which is larger than the length of 8CB molecules ($l = 1.8$ nm) and smaller than the length of pure 8CB smectic layers (3.2 nm)[52,55]. These values indicate that the smectic layers contain both 5CB and 8CB molecules. Interestingly, the first layer is significantly denser and more ordered than pure bulk 8CB in the smectic phase, by factors of 1.5 and 1.3, respectively.

**Simulation of pure 5CB.** In order to investigate the influence of composition on the features observed here for mixtures of 5CB/8CB, additional pure 5CB simulations are performed with the same cylinder size, 5 nm. As before, the sample consists of 16,000 5CB molecules, and the simulation is carried out at atmospheric pressure. A direct comparison between the 5CB/8CB mixture and pure 5CB can be established by performing the simulation at 295 K because the nematic–isotropic transition temperature for 5CB is 5 K lower than that of the mixture. Figure 4 shows the nematic director and order parameter calculated over the final 100 ns of simulation. The presence of the homeotropic cylinder breaks the symmetry and forms two line defects where the nematic director abruptly changes, as revealed by the low order parameter. The cylinder surface induces molecular order at the interface, forming high-order regions that follow an oscillatory pattern. However, these oscillations are less pronounced than the oscillations observed in the 5CB/8CB mixture (Fig. 1b). We also

calculate the biaxiality and density profile for pure 5CB, and we observed similar patterns to those reported for the mixture (Supplementary Note 8 and Supplementary Fig. 10).

**Mechanism of segregation.** One can determine the local concentration of 5CB and 8CB molecules by resorting to the following equation:

$$\frac{\rho_{8CB} - \rho_{5CB}}{\rho_{8CB} + \rho_{5CB}} \times 100 \qquad (1)$$

where $\rho_{8CB}$ and $\rho_{5CB}$ denote the 8CB and 5CB density, respectively. A positive value indicates a higher concentration of 8CB, while a negative value corresponds to a higher concentration of 5CB. In the bulk, the mixture is homogeneous (Supplementary Note 3). In the vicinity of surfaces and defects, pronounced deviations from the bulk concentrations are observed (Fig. 5). Around the cylinder, at the position of the first peak, the 8CB concentration is approximately 25% higher than that of 5CB. The longer 8CB molecules are localized in the region of high scalar order parameter, that is, the smectic layers. In contrast, the shorter 5CB molecules are localized in regions of low scalar order parameter. The antiparallel arrangement of 8CB molecules is energetically more favourable than that of 5CB molecules. In fact, at 300 K, pure 8CB adopts an antiparallel arrangement and a smectic phase in the bulk. Increasing the concentration of 8CB molecules in the smectic-like surface layers therefore reduces the energy of the system, leading to pronounced inhomogeneities whose origin is primarily enthalpic. To show that, we calculate the cohesive energy density, defined as the difference between the average potential energy of the gas phase and that of other, relevant condensed phases (in our case the nematic and smectic phases). For a small system consisting of 500 molecules at 300 K in the bulk, the cohesive energy densities are 15.49, 17.42 and 16.453 kcal mol$^{-1}$ for 5CB, 8CB and the 5CB/8CB mixture, respectively. As expected, the cohesive energy for 8CB in the smectic phase is higher than that of 5CB in the nematic phase, and the energy of a homogeneous 5CB/8CB mixture is an average of the 5CB and 8CB energies (a simple average would yield 16.459 kcal mol$^{-1}$). In the bulk, an equimolar 5CB/8CB mixture is thermodynamically stable and no phase separation occurs. That behaviour changes in the vicinity of an interface; when we prepare films for all three systems, with free interfaces (by increasing the size of the simulation box in one direction), the free surfaces induce two smectic layers on the two opposite sides of the simulation box. In this case, the cohesive energy for the mixture is 0.1 kcal mol$^{-1}$ higher than the average of the 5CB and 8CB cohesive energies, serving to emphasize that such compositionally inhomogeneous layering is energetically favourable. Our calculations thus show that when a surface induces smectic layers, an equimolar 5CB/8CB mixture can in fact decompose into two phases to increase cohesive energy. These findings can be generalized to the defects, and used to explain the localization of 5CB molecules in the defects where the scalar order parameter is low.

Additional insights into the segregation of 5CB/8CB mixture at the interface can be gained by performing additional simulations of a flat interface. The results of such simulations are discussed extensively in Supplementary Note 9. Here we merely point out that, as observed at the surface of the cylinder, 8CB molecules form smectic layers in the vicinity of the air interface, and exclude 5CB molecules from the layers. The concentration of 5CB and 8CB molecules follows a damped oscillatory pattern having the same frequency as that observed at the curved interface (Supplementary Fig. 11). These results reveal that the segregation of 8CB to high-order regions is induced by both curved and flat surfaces; we therefore conclude that the segregation observed in the defect is an intrinsic feature, and is not related to the

segregation that arises at the interface. Additional details pertaining to the simulation of a flat interface are provided in the Supplementary Note 9.

## Discussion

The results reported here provide a hitherto unavailable description of molecular segregation in experimentally relevant LC mixtures near surfaces and topological defects. In the defects of widely used mixtures of 8CB and 5CB, the local scalar order parameter is low and the biaxialty is large. The density and composition of the mixture in the defects are considerably different from those of the bulk material. The larger, 8CB molecules form smectic layers in the immediate vicinity of an interface. These layers tend to exclude the shorter 5CB molecules. This segregation is primarily due to enthalpic effects. In contrast, the density of the defects is lower than that of the bulk, and the local concentration of 5CB is significantly higher than the average. At the interface, LC molecules adopt an antiparallel structure to form quadrupoles. The results of our atomistic simulations are consistent with limited experimental observations and theoretical calculations in terms of continuum theories, which have anticipated the structure of defects on the basis of extrapolations to small length scales. Such theories or experiments, however, have not envisaged the compositional fluctuations discovered here, and their very existence offers intriguing possibilities for applications. Indeed, the fact that the local molecular environment of a defect differs appreciably from that of the bulk material raises new prospects for development of separation processes that rely on compositional segregation effects, for self-assembly, as revealed in recent work from our own groups[25,26], and for conducting chemical reactions, where density and orientational inhomogeneities could be manipulated to influence structure and yield.

## Methods

**Simulation protocol.** We perform MD simulations of 5CB/8CB mixture at the united atoms level of resolution[53]. The mixture sample consists of 8,000 5CB and 8,000 8CB molecules that are completely mixed at the initial configurations. The initial configurations are simulated in the NPT ensemble for 100 ns in the isotropic phase, $T = 320$ K, to fully equilibrate. Then, the systems are cooled down to $T = 300$ K and simulated for 200 ns to observe smooth transition from isotropic to nematic phase (Supplementary Fig. 1). All MD runs are carried out at atmospheric pressure, that is, $P = 101.3$ kPa, and the periodic boundary conditions are applied in all directions. To create a hole with cylindrical shape at the centre of simulation box a repulsive potential is applied as following:

$$U = K\left[1 + \cos\left(\frac{\pi r}{r_c}\right)\right] \quad r < r_c \qquad (2)$$

where $K$ is the potential constant, $r$ is the distance of atoms from the $Z$ axis and $r_c$ is the radius of the cylinder. The repulsive potential pushes all atoms located in the cylinder out and creates a hole with cylindrical shape. We run MD simulations with repulsive potential for 1,500 ns for equilibration and 100 ns for data analysis.

**Calculation of scalar order parameter and nematic director.** To investigate the orientation of molecules locally, the simulation box is divided into the small bins, and the local **Q** tensor is computed by averaging over all molecules present in each bin through the following expression:

$$\mathbf{Q} = \frac{1}{\sum_{t=0}^{T} N(t)} \sum_{t=0}^{T} \sum_{i=1}^{N(t)} \frac{1}{2}[3u_i(t) \otimes u_i(t) - I] \qquad (3)$$

where $u$ is the long molecular axis unit vector (Supplementary Fig. 2), $I$ is the identity matrix, $N$ is the number of molecules in each bin and $T$ is the simulation time. The size of each bin is adjusted to get a similar population of LC molecules ($N \sim 10$ in each bin). The local scalar order parameter in each bin is obtained as the largest eigenvalue of the **Q** tensor, and the corresponding eigenvector defines the nematic director. The more information about systems and calculation of **Q** tensor can be found in Supplementary Notes 1 and 2.

**Data availability.** All input files and scripts have been made available at https://github.com/m-rahimi/Defect_Scripts.git. Additional modified scripts can be accessed upon request.

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

## Acknowledgements

The design of triggerable materials based on LCs and their response to interfacial perturbations is supported by the US Army Research Office through the MURI program (W911NF-15-1-0568). The study of chemical structure and composition in topological defects is supported by NSF DMR-1410674. The simulations of structure and deformation of flat interfaces and the propagation of these effects into the bulk of a material are supported by NSF DMR-1420709. The study of self-assembly in topological defects is supported by NSF DMR-1121288. We gratefully acknowledge the computing resources

provided on Blues, a high-performance computing cluster operated by the Laboratory Computing Resource Center at Argonne National Laboratory. We further acknowledge the University of Chicago Research Computing Center (RCC) for allocation of computing resources.

## Author contributions

M.R., N.L.A. and J.J.d.P. designed the research; M.R., H.R.-D., A.R.-H. and J.J.d.P. performed the research; M.R., H.R.-D., A.R.-H. and J.J.d.P. analysed the data; R.Z. performed continuum theory calculations. M.R. and J.J.d.P. wrote the paper; N.L.A. and J.J.d.P. supervised the research. All authors discussed the progress of research and reviewed the manuscript.

## Additional information

**Competing interests:** The authors declare no competing financial interests.

**Publisher's note**: 

