## [Peer Review File · Nature Communications]

Reviewers' comments:

Reviewer #1 (Remarks to the Author):

A molecular view of disclination line defects in nematic liquid crystals

M. R., R. Zhang, H. Ramezani-Dakhel, A. Ramirez-Hernandez, N. L. Abbott, J.J. de Pablo

This paper presents an interesting atomistic molecular dynamics simulation study of line defects in a nematic liquid crystal mixture, showing details at molecular level of the defect core structure. The defects are created by inserting in the liquid crystal (a mixture of 8000 5CB and 8000 8CB molecules) a featureless vertical cylindrical rod endowed, with the help of an external field with a certain assumed radial decay law, of homeotropic aligning properties. This leads to the formation of two disclination line defects that run parallel to the cylinder.

A particularly interesting issue addressed for the first time is whether, for LC mixtures, the nematic director field defects have an influence on the local composition. The paper reports that this is the case and that the density within the defects is significantly lower than in the bulk.

My comments are:

1) The central cylinder is not treated at atomistic level, so it is difficult to see if the conclusions about the chemical composition selectivity can be generalized to chemically well defined nano-cylinders. A comment of this aspect would be welcome.

2) The claim that this is the first study at atomistic level of defect structure:

(Lines 82-83 "To the best of our knowledge, defects have never been analyzed with atomic-level resolution.") is too strong. I think this is true for line defects, but that a plane wall defect appearing in a very thin nano thin hybrid 5CB film as predicted by continuum theory [P. Palffy-Muhoray et al., 1994, *Liq. Cryst.*, 16, 713] was observed and characterized at atomistic level in [Pizzirusso, A. et al., 2012, *Chemical Science* 3: 573]. The same paper (not cited in the paper, but only in the supplementary information) also reports values for Rapini anchoring energy at an air interface (cf. line 125) and shows surface induced ordering also in the isotropic phase (cf. line 204).

3) (a) Lines 160-162 The statement "The biaxiality is given by the second largest eigenvalue of the Q tensor, representing molecular orientational fluctuations around the secondary director which is perpendicular to the nematic director" is not very precise and should be reformulated.

(b) Moreover, on a more technical level, an operational definition of the procedure used to evaluate the biaxiality plotted in fig.2b should be given at least in the S.I. (in particular mentioning the procedure used to order the Q eigenvalues so as to avoid overestimating the biaxiality).

(c) It would also be useful to have an estimate of the errors in density and order parameter as a function of distance from the surface (e.g. in fig.3). If the number of molecules in each bin is around 10 (line 279) an error of the order of 0.2-0.3 can be expected [see, e.g., Eppenga, R. and Frenkel, D. 1984, *Mol. Phys.* 52: 1303]. Is this the case?

4) Lines 223-226. The antiparallel arrangement of 5CB-5CB and 8CB-8CB pairs is described. Are the 5CB-8CB pairs also antiparallel?

5) Line 244-245 The statement "The results reported here provide a hitherto unavailable description of the molecular structure of realistic liquid crystalline materials near interfaces and defects" is probably too strong given that various atomistic simulations of 5CB near more realistic interfaces than the present have been reported.

In summary, this is a nice well written paper, by some of the leading authors in the field and collaborators. It reports original results for the molecular arrangement of the components of a nematic 5CB/8CB mixture at the core of the defects formed around a structureless cylinder immersed in the liquid crystal. I think it should be published, with minor corrections taking into account the points discussed above.

Reviewer #2 (Remarks to the Author):

"A molecular view of disclination line defects in nematic liquid crystals", M. Rahimi et.al.

I am not at all convinced that this paper is of general interest. The fact that the defect density is less than in the bulk and that there are small composition differences between the defect and the bulk is only of interest to a very small group of scientists. Furthermore, the claim that this can be used as a tool for assembly is only there to try and enhance the general appeal, but there is no focus on this aspect in the paper nor there is proof that this will actually be something one could use in any way in experiments. Hence, I recommend the paper is rejected by Nature Communications. From a technical point of view, I have no serious concerns. However, overall, I am not at all convinced that the results signify any significant scientific advancement.

Reviewer #3 (Remarks to the Author):

The article entitled « A molecular view of disclination line defects in nematic liquid crystals » is an interesting article that deserves publication in Nature Communication but major revision appears necessary to my opinion in order to clarify the information which is obtained from the simulation here presented.

First it would be necessary to determine if the observed phenomena are only related to the presence of a mixture of 5CB/8CB or if some of them are intrinsic features also characteristic of pure nematic compounds. The important features related to the defect are its size (the determination of the core size is also interesting and should be discussed according to the previously published theoretical and experimental results), the biaxiality and the density decrease within the defect. To know if these features are specific of a nematic disclination, comparison between 5CB/8CB mixtures and pure 5CB appears necessary. In the present version of the article the intrinsic nature of these features is not clearly specified and this is because the comparison is still lacking to my opinion. Possibly this simulation already exists but it is not straightforward according to the references proposed in the article. In a new simulation with pure 5CB with the same cylinder, size, biaxiality and density could be measured as well, allowing to confirm if the observed results are specific nematic features or only related to the presence of a mixture. Also the layering feature should be referred to the experimental x-ray reflectivity measurements performed with a number of molecules of the n-CB series (see for example PRE 77 (2008) p 031607)

Second, since the presence of the cylinders has two consequences, layering and defect formation, it would be important to decipher the relationship between the two features: is the 8CB concentration decrease in the defect related to the 8CB concentration increase in the layers and vice-versa? In order to clarify this point, the density in the layers formed at a similar but flat interface with the same 8CB/5CB mixture should be given (not only the cohesion energy). If the 8CB concentration values are similar in the layers formed at a flat interface and around the cylinder, it will be a real demonstration of the layering-induced segregation phenomenon. More importantly it will demonstrate as well that the observed inverse segregation in the defect is an intrinsic feature, not related to the close increasing of 8CB density that would be compensated in the defect.

When the comparison with these new simulations will be obtained, the article could be accordingly rewritten for a useful publication in Nature Communication. In particular abstract and titles should be changed since they are now misleading. The subject of the paper is currently not the nematic

defect structure but the behavior of a mixture in presence in the same time of a defect and a surface.

Reviewer #1

1) The central cylinder is not treated at atomistic level, so it is difficult to see if the conclusions about the chemical composition selectivity can be generalized to chemically well defined nano-cylinders. A comment of this aspect would be welcome.

We thank the Referee for raising this question. In our manuscript, the goal was to perturb the system as little as possible, thereby avoiding the additional effects that might be introduced by chemical complexity and specific interactions. In this first study, we therefore chose to introduce a completely “neutral” surface, whose influence is primarily topological and only affects packing and geometry. The Referee is correct – realistic chemical surfaces might affect the composition in additional ways. We do note that a nanometer interfacial region with high order parameter was also observed in simulations of 5CB at a silicon surface with planar anchoring [Chem. Sci. 3, 573 (2012)]. The results for that chemically-detailed system are consistent with those reported in our work: the silicon surface induces orientational and positional order, forming a region at the interface with higher order, consistent with our observation at the cylinder surface. A discussion of this issue has been included in the revised manuscript.

2) The claim that this is the first study at atomistic level of defect structure: (Lines 82-83 “To the best of our knowledge, defects have never been analyzed with atomic-level resolution.”) is too strong. I think this is true for line defects, but that a plane wall defect appearing in a very thin nano thin hybrid 5CB film as predicted by continuum theory [P. Palffy-Muhoray et al., 1994, Liq. Cryst., 16, 713] was observed and characterized at atomistic level in [Pizzirusso, A. et al., 2012, Chemical

Science 3: 573]. The same paper (not cited in the paper, but only in the supplementary information) also reports values for Rapini anchoring energy at an air interface (cf. line 125) and shows surface induced ordering also in the isotropic phase (cf. Line 204).

The reviewer is correct. In the atomistic simulation of a small hybrid channel [Chem. Sci. 3, 573 (2012)], the director field abruptly changed in the middle of the channel, forming a plane wall defect at the center. However, the presence of the wall defect wasn't mentioned by the authors, and the physics underlying the defect were not discussed explained. Following the Referee's suggestion, we have modified the sentence to read "To the best of our knowledge, line defects have never been analyzed with atomic-level resolution." in the revised version, which are highlighted in green. We have also included the Reference recommended by the Referee.

3) (a) Lines 160-162 The statement "The biaxiality is given by the second largest eigenvalue of the Q tensor, representing molecular orientational fluctuations around the secondary director which is perpendicular to the nematic director" is not very precise and should be reformulated. (b) Moreover, on a more technical level, an operational definition of the procedure used to evaluate the biaxiality plotted in fig.2b should be given at least in the S.I. (in particular mentioning the procedure used to order the Q eigenvalues so as to avoid overestimating the biaxiality). (c) It would also be useful to have an estimate of the errors in density and order parameter as a function of distance from the surface (e.g. in fig.3). If the number of molecules in each bin is around 10 (line 279) an error of the order of 0.2-0.3 can be expected [see, e.g., Eppenga, R. and Frenkel, D. 1984, Mol. Phys. 52: 1303]. Is this the case?

(a) We thank the Referee for pointing out this problem. We have rephrased the statement as follows: "Breaking the symmetry of molecular orientation along the nematic director orients the molecules along the biaxial direction, which is perpendicular to the nematic director. The biaxiality quantifies the degree of orientation along the biaxial direction and is given by the second largest eigenvalue of the Q tensor".

(b) Following the Referee's suggestion, we have explained the procedure to calculate biaxiality in the SI text.

(c) We thank the Referee for this question. The errors in density and order parameter have been calculated for all figures, and we have not observed any patterns in them as a function of distance. As noted by the Referee, the errors are less than 2% in both the nematic and isotropic phases. Here, we plotted the density and order parameter as a function of distance from the surface with error bars (fig 3b in the direction shown with a black arrow in fig 3a). The error bars show the standard deviations of the computed values over the final 100 ns of MD simulations. The size of the error bars in density and order parameter is smaller than the magnitude of the oscillations. We did not include

them in the figures for clarity. Following the Referee's suggestion, we have mentioned the range of the errors in the revised version, which are highlighted in green.

4) Lines 223-226. The antiparallel arrangement of 5CB-5CB and 8CB-8CB pairs is described. Are the 5CB-8CB pairs also antiparallel?

We thank the Referee for raising this question. We address it here in two steps. First, we have calculated the radial distribution function (RDF) between the 5CB-5CB, 8CB-8CB, and 5CB-8CB pairs in the bulk of the 5CB/8CB mixture. The pair distance is defined as the distance between the centers of two vectors describing the direction of the molecules shown in Figure 2 SI. The RDFs show that the average number of 5CB and 8CB molecules in the neighborhood of a given molecule is nearly identical, indicating that the pairs have similar arrangements and the mixture is homogeneous. In the next step, in order to address the

antiparallel arrangement of molecules, we calculated the dot product between two vectors defining the direction of two molecules whose pair distances are smaller than 0.5 nm, where the first peak occurs in the RDFs. The sum of the dot products over all pairs measures the arrangement of the molecules; a negative value shows antiparallel arrangement, while a positive value shows parallel arrangement. We measured the sum of the dot products over all pairs whose pair distances are smaller than 0.5 nm in the bulk of the mixture, and we obtained -0.354, -0.357, and -0.356 for 5CB-5CB, 8CB-8CB, and 5CB-8CB pairs, respectively.

5) Line 244-245 The statement “The results reported here provide a hitherto unavailable description of the molecular structure of realistic liquid crystalline materials near interfaces and defects” is probably too strong given that various atomistic simulations of 5CB near more realistic interfaces than the present have been reported.

Following the Referee's suggestion, we have modified the sentence to “The results reported here provide a hitherto unavailable description of molecular segregation near surfaces and topological defects”

Reviewer #2

I am not at all convinced that this paper is of general interest. The fact that the defect density is less than in the bulk and that there are small composition differences between the defect and the bulk is only of interest to a very small group of scientists. Furthermore, the claim that this can be used as a tool for assembly is only there to try and enhance the general appeal, but there is no focus on this aspect in the paper nor there is proof that this will actually be something one could use in any way in experiments. Hence, I recommend the paper is rejected by Nature Communications. From a technical point of view, I have no serious concerns. However, overall, I am not at all convinced that the results signify any significant scientific advancement.

We appreciate the Referee's comments and his/her perspective. We respectfully disagree with the first comment in that liquid crystalline systems are widely used in displays, and are garnering renewed interest in chemical sensing, biophysics, and functional materials (ranging from optomechanical materials to chemomechanical actuators). We also disagree with the comment that there is no evidence nor proof that the topological defects can be used as a tool for self-assembly. Topological defects have been widely used to organize nanoparticles and polymerize mixtures of monomers. Recent experiments from our own work have in fact shown that defects can be used as templates for molecular self-assembly [Nat. Mater. 15, 106 (2016)]. We note that such a study has been highly cited, serving to underscore the general interest in the study of defects. In that work, the nanoscopic environment created by topological defects in 5CB was used to trigger the self-assembly of molecular amphiphiles. In another experimental study, it was shown that the self-assembly of amphiphiles into topological defects strongly depends on the structure of the defects [Phys. Rev. Lett. 116, 147801 (2016)]; self-assembly was observed into a +1 point defect at low concentrations of amphiphile (47 μM), while it was observed into a -1 point defect at high concentrations of amphiphile (570 μM). In an ongoing project, we are using the structure of defects presented in this paper to study molecular self-assembly. Our preliminary results for one amphiphile show a minimum in the potential of mean force close to the defect, where the amphiphile interacts with the defect by inserting its hydrophilic head into it, while exposing the hydrophobic tail to the nematic phase, consistent with experimental observations. Based on our studies and observations, we therefore believe that composition differences between the defect and the environment around the defect are key to understanding molecular self-assembly in defects. It is worth mentioning that these simulations are extraordinarily demanding; we expect to complete them in approximately one year, and hope to publish the results in the future.

Reviewer #3

First it would be necessary to determine if the observed phenomena are only related to the presence of a mixture of 5CB/8CB or if some of them are intrinsic features also characteristic of pure nematic compounds. The important features related to the defect are its size (the determination of the core size is also interesting and should be discussed according to the previously published theoretical and experimental results), the biaxiality and the density decrease within the defect. To know if these features are specific of a nematic disclination, comparison between 5CB/8CB mixtures and pure 5CB appears necessary. In the present version of the article the intrinsic nature of these features is not clearly specified and this is because the comparison is still lacking to my opinion. Possibly this simulation already exists but it is not straightforward according to the references proposed in the article. In a new simulation with pure 5CB with the same cylinder, size, biaxiality and density could be measured as well, allowing to confirm if the observed results are specific nematic features or only related to the presence of a mixture.

We thank the Referee for raising this question. To address this comment, we performed extensive, additional pure-5CB simulations with the same cylinder size, 5 nm. As for the mixture, the sample consists of 16000 5CB molecules, and the simulation is carried out at atmospheric pressure. To make a direct comparison between the 5CB/8CB mixture and pure 5CB, we performed the simulation at 295 K, since the nematic-isotropic transition temperature for 5CB is 5 K smaller than that for the mixture. The figure shows the simulated nematic director and order parameter averaged over 100 ns. As for the mixture, the presence of the homeotropic cylinder breaks the symmetry and forms two line defects where the nematic director abruptly changes, as revealed by the low order parameter. The cylinder surface induces molecular order at the interface, forming high-order regions that follow an oscillatory pattern. However, these oscillations are less pronounced than the oscillations observed in the 5CB/8CB mixture (fig 1b). We also calculated the biaxiality and density profile for pure 5CB, and we observed similar patterns as those reported for the mixture. These results and the figures showing the nematic director and order parameter for pure 5CB have been added to the revised version of the manuscript.

The figure below shows the order parameter, biaxiality, and density in polar coordinates for pure 5CB. The value of the order parameter exceeds the bulk value in the region close to the surface (Fig. a). In contrast to the mixture, the second and third peaks in the order parameter are not clearly defined, showing that the order parameter oscillations damp faster in the pure system than in the mixture. The defects are localized after the first peak, and they reduce the order parameter in the first peak. The calculated biaxiality for pure 5CB shows a non-zero value in small regions of space localized at the defects (Fig. b). Fig. c shows the oscillation pattern for the density profile, indicating that 5CB molecules form organized layers around the cylinder surface. In the first layer 5CB molecules possess both orientational and positional order, forming a smectic layer. This observation is particularly interesting, since pure 5CB molecules in the bulk cannot form a smectic phase. Fig. d shows the oscillation in the order parameter and density along two different directions shown in Fig. c. In the direction going through the bulk, shown with black color, we observe similar damped oscillations in the order parameter and density; the number and frequency of the maxima in both curves are the same. In contrast, in the direction going through the defect, the order parameter is significantly lower than the bulk value, and the density is lower than in other regions at the same distance from the cylinder surface.

Also the layering feature should be referred to the experimental x-ray reflectivity measurements performed with a number of molecules of the n-CB series (see for example PRE 77 (2008) p 031607)

We thank the Referee for suggesting a reference. We have added the reference to the revised version.

Second, since the presence of the cylinders has two consequences, layering and defect formation, it would be important to decipher the relationship between the two features: is the 8CB concentration decrease in the defect related to the 8CB concentration increase in the layers and vice-versa? In order to clarify this point, the density in the layers formed at a similar but flat interface with the same 8CB/5CB mixture should be given (not only the cohesion energy). If the 8CB concentration values are similar in the layers formed at a flat interface and around the cylinder, it will be a real demonstration of the layering-induced segregation phenomenon. More

importantly it will demonstrate as well that the observed inverse segregation in the defect is an intrinsic feature, not related to the close increasing of 8CB density that would be compensated in the defect.

We thank the Referee for raising this question. To study the phase separation at a flat interface, we have carried out additional simulations of 4000 5CB and 4000 8CB molecules at 300 K. The initial configuration is generated with the same method explained in detail in the SI text. The system is fully equilibrated in a cuboid simulation box with periodic boundary conditions, and the equilibrated box has dimensions $11 \times 11 \times 30$ nm³. Then, we increase the simulation box in the Z direction to sandwich the 5CB/8CB mixture between vacuum, thereby forming a film, and run the simulation in the NVT ensemble. Molecules form two flat surfaces with homeotropic anchoring at the two opposite sides of the film. We perform the MD simulation for 400 ns for equilibration and 100 ns for data collection.

Figure shows the concentration of 5CB and 8CB molecules as a function of distance for the system with the cylinder and the system with a flat interface. We excluded the defects from our calculations for systems with a cylinder. The concentration of 5CB and 8CB molecules follows a damped oscillatory pattern, and continue over a length scale of almost 10 nm from the surface. Three maxima in the concentration of 8CB are visible for both systems. For the system with a cylinder, only three maxima are visible at 1.6, 4.3, and 6.9 nm from the surface, which correspond to the three maxima in the order parameter (Fig 3b).

For the system with a flat interface, the three maxima shift to the left but their frequency is identical to that observed in the system with a cylinder. These results reveal that the segregation of 8CB in the high order region is induced by both curved and flat surfaces. In both cases, 8CB molecules form smectic layers in the vicinity of the interface and exclude 5CB molecules from the layers, leading to the segregation phenomenon reported in the manuscript. It is worth mentioning that the magnitude of 8CB segregation in the layers is slightly more pronounced in the system with the cylinder. Note that the phase separation typically happens on microsecond length scales. Here, our long timescale MD simulations relied on a high-performance computing cluster operated by the Laboratory Computing Resource Center at Argonne National Laboratory and required more than one year.

When the comparison with these new simulations will be obtained, the article could be accordingly rewritten for a useful publication in Nature Communication. In particular abstract and titles should be changed since they are now misleading. The subject of the paper is currently not the nematic defect structure but the behavior of a mixture in presence in the same time of a defect and a surface.

REVIEWERS' COMMENTS:

Reviewer #1 (Remarks to the Author):

I have read the reply letter to my previous comments and the revised version of the manuscript and I am satisfied with the changes made.

Thus in my opinion the manuscript can now be published in Nat. Comm.

Reviewer #3 (Remarks to the Author):

I consider that most of the comments have been taken into account and I suggest to publish the paper